Local norms of cheating and the cultural evolution of crime and punishment: a study of two urban neighborhoods

Schroeder Kari Britt 1 2 kari.britt.schroeder@gmail.com
Pepper Gillian V. 1
Nettle Daniel 1
1 Centre for Behaviour and Evolution, Newcastle University , Newcastle Upon Tyne , United Kingdom
2 Department of Psychological and Brain Sciences, Boston University , Boston, MA , United States of America
Roberts David
Electronic publication date: 2014 Jul 1
Publication date: 2014
Volume: 2
Electronic Location ID: e450
Received 2014 Mar 25; Accepted 2014 Jun 4
Copyright: © 2014 Schroeder et al.
Copyright year: 2014
Copyright holder: Schroeder et al.
License: This is an open access article distributed under the terms of the Creative Commons Attribution License, which permits unrestricted use, distribution, reproduction and adaptation in any medium and for any purpose provided that it is properly attributed. For attribution, the original author(s), title, publication source (PeerJ) and either DOI or URL of the article must be cited.
License URL: https://creativecommons.org/licenses/by/4.0/

Keywords: Social disorganization theory, Cooperation, Descriptive norms, Injunctive norms, Social capital, Cultural evolution

Funding: NSF Award #1003961 Funding for this study was provided by National Science Foundation Award #1003961 to KB Schroeder. The funders had no role in study design, data collection and analysis, decision to publish, or preparation of the manuscript.

==============================
The prevalence of antisocial behavior varies across time and place. The likelihood of committing such behavior is affected by, and also affects, the local social environment. To further our understanding of this dynamic process, we conducted two studies of antisocial behavior, punishment, and social norms. These studies took place in two neighborhoods in Newcastle Upon Tyne, England. According to a previous study, Neighborhood A enjoys relatively low frequencies of antisocial behavior and crime and high levels of social capital. In contrast, Neighborhood B is characterized by relatively high frequencies of antisocial behavior and crime and low levels of social capital. In Study 1, we used an economic game to assess neighborhood differences in theft, third-party punishment (3PP) of theft, and expectation of 3PP. Participants also reported their perceived neighborhood frequency of cooperative norm violation (“cheating”). Participants in Neighborhood B thought that their neighbors commonly cheat but did not condone cheating. They stole more money from their neighbors in the game, and were less punitive of those who did, than the residents of Neighborhood A. Perceived cheating was positively associated with theft, negatively associated with the expectation of 3PP, and central to the neighborhood difference. Lower trust in one’s neighbors and a greater subjective value of the monetary cost of punishment contributed to the reduced punishment observed in Neighborhood B. In Study 2, we examined the causality of cooperative norm violation on expectation of 3PP with a norms manipulation. Residents in Neighborhood B who were informed that cheating is locally uncommon were more expectant of 3PP. In sum, our results provide support for three potentially simultaneous positive feedback mechanisms by which the perception that others are behaving antisocially can lead to further antisocial behavior: (1) motivation to avoid being suckered, (2) decreased punishment of antisocial behavior, and (3) decreased expectation of punishment of antisocial behavior. Consideration of these mechanisms and of norm psychology will help us to understand how neighborhoods can descend into an antisocial culture and get stuck there.

Introduction

Why do humans behave antisocially? The converse of this question—why humans behave prosocially—has been studied extensively by experimental economists, and determinants of prosocial behavior may be mirror images of determinants of antisocial behavior. One proximate explanation for prosocial behavior is punishment; i.e., people will behave prosocially if not doing so results in punishment. Empirical evidence for this comes from economic games. Using a repeated public goods game, Yamagishi (1986) and Fehr & Gächter (2000) showed that the opportunity for players to fine each other on the basis of contribution behavior can stabilize contributions to the public good at a high level. Following this, the cross-cultural covariation of prosocial behavior and punishment has received substantial interest (Henrich et al., 2006; Herrmann, Thöni & Gächter, 2008). Considerable local variation in prosociality has also been observed (Wilson, O’Brien & Sesma, 2009; Nettle, Colléony & Cockerill, 2011; Lamba & Mace, 2011), yet the question of whether prosocial behavior and punishment positively covary at the local level has spurred little research among experimental economists (but see Kocher, Martinsson & Visser, 2012).

However, the related question of whether antisocial behavior and a lack of punishment positively covary at the neighborhood level has generated substantial research within the field of sociology. Social disorganization theory posits that poverty, residential mobility, and family disruption can diminish the capacity a community has for creating relationships and establishing shared social norms. This low level of ‘social capital’ can lead to increased crime and delinquency via reduced collective action (Shaw & McKay, 1942; Sampson & Groves, 1989; Bursik & Grasmick, 1993a; Bursik & Grasmick, 1993b; Sampson, Raudenbush & Earls, 1997). Without trust and shared behavioral expectations, residents have decreased capacity to enforce desirable behavior through informal social control (i.e., informal surveillance and/or intervention by residents) (Sampson, Morenoff & Gannon-Rowley, 2002).

Of interest to researchers in both of these fields is how the local social environment can evolve over time to become more prosocial or more antisocial. This requires an understanding of the dynamic relationship between individual decisions (as typically studied by experimental economists) and the local social environment (as typically studied by sociologists). That is, individual decisions can be influenced by empirical expectations of the behavior of others in the local social environment (Bichierri & Xiao, 2009). These decisions, as manifest in observable behavior, then become part of the local social environment. Others will form expectations on the basis of their perception of the local environment and possibly alter their own behavior. That such a dynamic relationship exists is suggested by, for example, the interdependence of individual decisions to commit crimes (Glaeser, Sacerdote & Scheinkman, 1996).

In this paper, we attempt to bridge these two approaches of experimental economics and sociology and increase our understanding of the dynamic relationship between individual decisions and the social environment. We do so through consideration of the role of the individual’s expectation of others’ cooperative behavior—that is, the role of perceived local norms of cooperative behavior. Cialdini, Reno & Kallgren (1990) distinguish between injunctive norms and descriptive norms. Injunctive norms convey how people should behave. Descriptive norms, on the other hand, illustrate how most people actually do behave.

It is readily apparent that cooperative descriptive norms should be informative as to people’s expectation of cooperation. However, cooperative descriptive norms may also be informative as to people’s expectation of punishment for cooperative norm violation or antisocial behavior, particularly when there is a mismatch between injunctive and descriptive norms. A lack of alignment between injunctive and descriptive cooperative norms is implicit in broken windows theory—the idea that signs of social and physical disorder invite criminal behavior—in part because disorder is a cue that social control is lax (Kelling & Wilson, 1982). This mechanism for the ‘spread of disorder’ was elegantly tested by Keizer, Lindenberg & Steg (2008), who created public spaces in which an explicit injunctive norm was violated—e.g., a littered space (conveying a descriptive norm) next to a sign telling people not to litter (injunctive norm)—thereby communicating a lack of adherence to the injunctive norm and experimentally inducing further antisocial behavior. These results suggest that signs that others are flouting injunctive cooperative norms may serve as cues that antisocial behavior will not be punished. However, this remains a largely untested explanation of these results and of the broken windows effect in general (Traxler & Winter, 2012; but see Lochner, 2007).

Important to the studies we present in this paper, the work of Keizer, Lindenberg & Steg (2008) and Keizer, Lindenberg & Steg (2013) also demonstrated the possibility for ‘cross-norm effects’—that is, the focus of the injunctive and descriptive norms was different from the behavioral outcome assessed by the researchers. Some of the observed cross-norm effects included public versus private goods. For example, graffiti and litter (destruction of a public good) each resulted in an increase in theft of an envelope with money in it (Keizer, Lindenberg & Steg, 2008). In another set of experiments, these same authors also demonstrated cross-norm effects for the restoration of a public good and prosocial behavior targeted at an individual; garbage bags on the street—in violation of city ordinance—resulted in a decrease in posting of a letter dropped next to a postbox (Keizer, Lindenberg & Steg, 2013).

Thus, studying injunctive and descriptive cooperative norms presents a way to assess individual perceptions of environmental variation in cooperative and, potentially, punitive behavior. It also offers a way to study how the social environment affects the behavior of the individual and individual’s behavior in turn affects the social environment, by conveying information about descriptive norms. It is particularly appropriate when the focus is on local (rather than large-scale) variation in prosocial or antisocial behavior, as injunctive norms may be more similar in areas where people share common culture and history, while descriptive norms may still vary. Given a general consensus on injunctive norms, the emphasis can then be on perceived deviation from the injunctive norms.

The studies

Our studies were set in two nearby neighborhoods in Newcastle Upon Tyne, England, that we expected to have similar injunctive cooperative norms based on a shared cultural history. These two neighborhoods are similar in size, physical layout, and ethnic composition yet differ dramatically in rates of antisocial behavior and socioeconomic deprivation. While Neighborhood A is relatively affluent, Neighborhood B has experienced high rates of unemployment, physical decay, massive depopulation, and crime, following the collapse of mining and shipbuilding industries (see Nettle, Colléony & Cockerill, 2011 and citations therein). In an earlier study, Nettle, Colléony & Cockerill (2011) used surveys, a Dictator Game, behavioral observation, and field experiments to reveal substantially less antisocial behavior, more social capital, and more prosocial behavior in Neighborhood A than B.

Here, we return to these neighborhoods to investigate whether individual decisions to engage in antisocial behavior and norm enforcement vary by neighborhood. To do so, we evaluated antisocial behavior, punishment, and expectation of punishment in an economic game. We used a questionnaire to investigate whether neighborhood differences in antisocial behavior, punishment, and expectation of punishment could be explained by neighborhood differences in trust and local descriptive cooperative norms. Study 1 was observational and aimed to document and explain differences in perceptions and behaviors between the neighborhoods. Study 2 introduced a novel experimental methodology to manipulate perceived injunctive norm adherence, allowing us to make causal inferences. We assessed whether information on injunctive cooperative norm adherence altered expectations of punishment for antisocial behavior.

Study 1

Camerer & Fehr (2004) suggest that a real-world example of a third-party punishment game (3PP game) (Fehr & Fischbacher, 2004) is scolding of a neighbor for treating another person unacceptably. In this study, we administered a 3PP game along with a questionnaire (see Supplemental Information). Our variant of the game, which was played among residents within each neighborhood, enabled us to study differences between the neighborhoods in antisocial behavior and punishment for antisocial behavior. Player 1 was given the opportunity to steal from Player 2. Player 3 was given the opportunity to fine Player 1 if she took money from Player 2. Player 2 indicated whether she thought Player 3 would fine Player 1 if Player 1 took half of Player 2’s money.

We used Player 1 and Player 2 decisions to assess whether residents of Neighborhood B were (1) more likely to behave antisocially and (2) less likely to expect someone in their neighborhood to intervene in antisocial behavior. In conjunction with the questionnaire, we also used Player 1 and 2 decisions to investigate (3) whether perceived local cooperative norm violation could explain the the hypothesized neighborhood differences in individual antisocial decisions and (4) punitive expectations.

We used Player 3 game decisions and the questionnaire to assess (1) whether residents of Neighborhood A were more willing than those of B to punish antisocial behavior in their neighborhood, and (2) whether, following social disorganization theory, neighborhood trust could explain the hypothesized relationship between neighborhood and punitive behavior.

Study 1 methods

Sampling

The Ethics Committee of the Newcastle University Faculty of Medical Sciences approved the study protocol (Protocol #00503/2011 and Amendment #00503_1/2012). Written consent was not obtained because it would have been the only record of participation. We conducted the study from July 2012 to December 2012. A maximum of one participant per household was drawn from the electoral roll. Potential participants received a hand-delivered envelope with a cover letter describing the study, packet (questionnaire, explanation of the game, and game), and stamped return envelope. A minority of envelopes were delivered by subjects in another study (Nettle et al., 2014). We avoided sampling adjacent households and households sampled by Nettle, Colléony & Cockerill (2011).

Questionnaire

From the questionnaire, we recorded each participant’s age and sex.

Trust

We asked individuals how much they trust people in their neighborhood, on a 10-point scale (10 = most trusting).

Civic norms: condoned and perceived cheating

We asked individuals about both injunctive and descriptive civic norms (Supplemental Information). For the injunctive norms, we described three behaviors and asked whether it is Never OK to do this behavior, Always OK, or somewhere in between. Answers were constrained to a 10-point scale (1 = ‘Always OK’ and 10 = ‘Never OK’). The behaviors were (1) cheating the benefits system, (2) avoiding a fare on public transport, and (3) cheating on taxes. Condoned cheating is the average across behaviors. Larger values indicate that cheating on public goods is condoned. Note that condoned cheating is similar to the ‘norms of civic cooperation’ (Knack & Keefer, 1997; Herrmann, Thöni & Gächter, 2008) derived from the World Values Survey.

For the descriptive norms, we asked individuals whether they think many people in their neighborhood would do these behaviors (1 = ‘No one would’ and 10 = ‘Everyone would’). We averaged across these responses to arrive at perceived cheating. Larger values indicate that neighborhood cheating on public goods is perceived as more common.

We note that the cooperative norms used in the questionnaire pertain to public goods, while the possibility for antisocial behavior in the game is directed at a single person. However, as mentioned in the introduction, previous studies have experimentally demonstrated ‘cross-norm effects’ wherein destruction or restoration of a public good induced antisocial or prosocial behavior, respectively, directed at a single individual (Keizer, Lindenberg & Steg, 2008; Keizer, Lindenberg & Steg, 2013).

The 3PP game

Participants read instructions for the game, which followed the questionnaire, and then worked through examples (see Supplemental Information). (From this, we had responses to six test questions.) They were told that after receiving the packet in the post, we would determine the game outcome and then deliver their cash payoff along with a £5 payment for completing the survey.

The game worked as follows: all three players received an initial allocation of £10, to be paid after the decisions of all three players had been submitted. Player 1 had to decide how many pounds (integer from 0 to 10) to take from Player 2. If Player 1 took money from Player 2, Player 3 had to decide whether to fine Player 1. We used the strategy method for Player 3. Player 3 had to decide, for each amount greater than 0 that Player 1 could take, whether to pay to fine Player 1. Therefore, Player 3 had to make 10 choices, each corresponding to an amount that Player 1 might take from Player 2. The cost of the fine to Players 1 and 3 was constant (Player 3 paid £2 to make Player 1 lose £6). Player 2 could not make a choice in the game. We asked Player 2 to indicate whether she thought Player 3 would fine Player 1 if she took £5 from her (Supplemental Information).

Game behaviors are thus: theft (an integer from 0 to 10 representing the amount of money Player 1 took from Player 2), expect 3PP (whether Player 2 expected Player 3 to punish Player 1 if she took £5), and punitiveness (an integer from 0 to 10; this is the total number of potential thefts, from £1–£10, that Player 3 would punish).

Subjective value of money

We expected the subjective value of money to differ between neighborhoods and impact game behavior. Therefore, following the game, we asked how much of a difference, on a scale of one to 10, an amount of money x would make to their weekly budget, where x was £1 for Player 1 (value £1) and £2 for Players 2 and 3 (value £2). After commencement of data collection, we revised the packets for Player 1 to include x = £10. Thus, for some Player 1s we also have value £10.

Statistical analyses

The majority of responses can be considered discrete ordered choices. Thus, to assess neighborhood differences in game behavior, trust, cooperative norms, and the value of money, we analyzed the data with ordered logistic regression. The exception to this is game behavior for Player 2, for which we used binary logistic regression. We compared the fit of different models with the Akaike information criterion (AIC) (Akaike, 1974). Ordered and binomial logistic regression analyses and plotted predictions (i.e., the predicted value based on the fitted model and the data used to fit the model) were produced in the R statistical and computing environment (R Core Team, 2012) with the following packages: MASS (Venables & Ripley, 2002), rethinking (McElreath, 2012), beeswarm (Ecklund, 2012), and ggplot2 (Wickham, 2009). Note that plotted predictions for theft and punitiveness are both (0, 8). For each of these game behaviors, two possible values were not observed (3 and 8 for theft, 2 and 9 for punitiveness); thus, for prediction we condensed the ranges. We report Odds Ratios (ORs) for a unit increase in the outcome for each unit increase of the predictor variable, accompanied by 95% confidence intervals.

Study 1 results

Participants

We achieved sample sizes of 40 (16 male), 44 (22 male), and 49 (23 male) for Players 1, 2, and 3, respectively, in Neighborhood A and 34 (12 male), 43 (23 male), and 50 (23 male) in B (Table S1). Every week, new players from each neighborhood were combined into triads, and we determined game outcome from their decisions. For incomplete triads, players were drawn at random from all previous neighborhood players. We delivered to participants: the game outcome, debriefing sheet, money received from the game, and £5 for participating. The mean payoff from the game is £9.26 (σ = £3.49) in Neighborhood A and £9.16 in B (σ = £4.13). Descriptive statistics and neighborhood comparisons for key variables are in Table 1. We also report descriptive statistics in the text to assist the reader.

Table 1 Key variables from Study 1 by neighborhood.

Columns one and two contain medians for Neighborhood A and Neighborhood B, respectively (median absolute deviation in parentheses). Column three contains the odds that a participant from Neighborhood B indicated a higher value (95% confidence interval in parentheses). Condoned cheating and perceived cheating are the mean of the three injunctive and descriptive norms, respectively.

Variable	Median A (scale 1 to 10)	Median B (scale 1 to 10)	Odds B higher	
Trust neighbors	8 (1)	5 (2)	0.053 (0.031, 0.093)	
Value £1	1 (0)	1 (0)	1.89 (0.77, 4.61)	
Value £2	1 (0)	2.5 (1.5)	5.53 (2.51, 12.18)	
Value £10	3 (1)	5 (3)	3.37 (1.04, 10.9)	
Injunctive avoid fare	1 (0)	1 (0)	0.84 (0.52, 1.35)	
Injunctive cheat benefits	1 (0)	1 (0)	1.43 (0.87, 2.34)	
Injunctive cheat tax	1 (0)	1 (0)	1.38 (0.84, 2.23)	
Descriptive avoid fare	3 (1)	6 (2)	11.02 (6.58, 18.46)	
Descriptive cheat benefits	3 (1)	6 (2)	13.95 (8.16, 23.85)	
Descriptive cheat tax	4 (1)	5 (2)	3.06 (1.95, 4.79)	
Condoned cheating	1.33 (0.33)	1.50 (0.50)	1.25 (0.80, 1.95)	
Perceived cheating	3.00 (0.67)	5.50 (1.83)	10.22 (6.18, 16.90)	

Trust

Participants in Neighborhood A indicated far higher trust neighbors (median 8 on a scale of 1:10, median absolute deviation (MAD) 1) than did participants in B (median 5, MAD 2) (Table 1) (OR 18.8, 95% CI [10.8–32.8]).

Punishment of antisocial behavior

As predicted, participants in Neighborhood A were more punitive than those in B (Fig. 1) (OR 3.3, 95% CI [1.6–7.0]). Median punitiveness is 6 (MAD 4) and 3 (MAD 3) for Neighborhoods A and B, respectively. Thus, more participants in Neighborhood A indicated that they would pay £2 to fine Player 1 for a greater number of potential thefts.

Figure 1 Player 3 behavior, punitiveness, by neighborhood.

Each diamond represents one observation.

The subjective cost of punishment in the game, value £2, had a negative effect upon punitiveness (OR 0.7, 95% CI [0.6–0.9]) and was larger for participants in Neighborhood B than A (Table 1). However, participants in Neighborhood A were still more punitive than those in B when we include value £2 in the model (OR 2.1, 95% CI [0.9–4.6]). This result is robust to the inclusion of additional covariates age, male, and test questions (OR 2.9, 95% CI [1.2–7.2]).

Based on social disorganization theory, we hypothesized that greater trust among residents of Neighborhood A would partially explain the increased willingness of residents to engage in 3PP of antisocial behavior. Individuals who reported greater trust neighbors were slightly more punitive (OR 1.15, 95% CI [0.99–1.32]). The relationship between trust and punitiveness is less robust to the inclusion of value £2 (OR 1.09, 95% CI [0.94–1.27]); however, including an interaction between value £2 and trust neighbors improves model fit (AIC of 380.13 compared to 384.49).

Predictions from this model including the interaction are shown in Fig. 2. Value £2 still has a negative effect on punitiveness, but the slope is steeper for participants with high trust neighbors. Thus, participants with high trust neighbors are more punitive than those with low trust neighbors when value £2 is small, but less punitive when it is large. Neighborhood is no longer a reliable predictor of punitiveness when the interaction is included in the model (OR 1.8, 95% CI [0.7, 5.7]), nor does model fit improve with the addition of neighborhood (AIC = 380.67).

Figure 2 Punitiveness modeled as an interaction between trust neighbors and value £2.

Blue is ‘high trust’ (8; median trust neighbors score for Neighborhood A). Orange is ‘low trust’ (5; median trust neighbors score for Neighborhood B). Dotted lines are 95% confidence intervals.

Civic norms: condoned and perceived cheating

In both neighborhoods, most participants indicated that it is not acceptable to cheat on public goods. We observed little variation in injunctive norms across cooperative behaviors (Table 1). Nor did we detect a clear difference between neighborhoods with respect to specific injunctive norms or condoned cheating (i.e., the within-participant mean of injunctive norms) (Fig. 3, Table 1).

Figure 3 Neighborhood means and standard errors for condoned cheating and perceived cheating.

For condoned cheating: 1, Never OK; 10, Always OK; and for perceived cheating, 1, No one would; 10, Everyone would.

However, there was a dramatic difference between neighborhoods with respect to perceived cheating. Participants in Neighborhood B indicated that more of their neighbors would cheat on a public good than those in A (median 3.00, MAD 0.67 for A; median 5.50, MAD 1.83 for B) (Fig. 3, Table 1). Participants who thought more of their neighbors cheat on public goods were also less trusting of their neighbors (OR 0.54, 95% CI [0.48–0.62]).

Juxtaposition of condoned cheating and perceived cheating reveals that although participants in Neighborhood B tended to state that many of their neighbors cheat on public goods, we lack strong evidence that they view this behavior as more acceptable than those in A. This fits with our prior expectation that injunctive cooperative norms would be similar in Neighborhoods A and B. We therefore use perceived cheating as a within-participant measure of perceived local cooperative norm violation, or deviation from the injunctive cooperative norm.

Antisocial behavior

Participants in Neighborhood B took more from their neighbors in the game. Theft is also more variable in Neighborhood B than A. The median value of theft is 5 in Neighborhood B (MAD 5), compared to 0 in A (MAD 0) (odds that theft is greater in Neighborhood B: OR 2.9, 95% CI [1.2–7.1]). The neighborhood difference in theft is robust to the inclusion of age, male, and value £1 (OR 2.8, 95% [2.5–6.9]). For the reduced dataset for which we had data on value £10 (40 participants, 23 from Neighborhood A), substituting this variable in the model increases the odds that a participant in B stole more in the game (OR 4.1, 95% CI [0.9–17.5]). Inclusion of test questions in the model reduces confidence in the neighborhood difference in theft (OR 2.1, 95% [0.8–5.8]). However, incomplete test questions are heavily patterned for Player 1; only participants in Neighborhood B for whom theft > 0 did not complete the questions. Irrespective of the participant’s comprehension of the entire game, the opportunity for Player 1 to behave antisocially (the outcome of interest to us) should be very clear from the packet (i.e., “How many pounds do you choose to take from Player B?”) (Supplemental Information).

As expected, perceived cheating is a robust predictor of theft, even controlling for value £1 (Fig. 4; OR 1.3, 95% CI [1.0–1.6]). When both neighborhood and perceived cheating are considered in the same model, neither is a reliable predictor of theft. Nor does AIC offer strong support for a single model (235.40 for the model with perceived cheating, 234.67 for neighborhood, and 234.60 for perceived cheating + neighborhood). This suggests that to understand the greater theft in Neighborhood B, we need to consider perceived cheating.

Figure 4 Theft for Player 1 modeled as dependent on perceived cheating.

Dotted lines are 95% confidence intervals. Bubbles represent the actual data from Neighborhood A (blue) and Neighborhood B (orange). Size of the bubble corresponds to the number of observations.

Expectation of 3PP

We asked Player 2 whether she thought Player 3 would fine Player 1 if Player 1 took £5 from her (expect 3PP). Contrary to our expectations, neighborhood was not a reliable predictor of expect 3PP. Of participants in Neighborhood A, 36.36% expected 3PP, compared to 30.23% of participants from Neighborhood B (OR 1.2, 95% CI [0.5–3.2]). However, as predicted, we did observe a negative relationship between perceived cheating and expect 3PP (Fig. 5; OR 0.8, 95% CI [0.6–1]). This relationship does not change with inclusion of value £2 as a proxy for the local subjective value of £2 (OR 0.8, 95% CI [0.6–1]).

Figure 5 Probability of expect 3PP dependent on perceived cheating.

Dotted lines are 95% confidence intervals.

Study 1 summary and discussion

Study 1 reveals that individual perceptions of local cooperative descriptive norms (i.e., perceived cheating) vary dramatically by neighborhood, in concordance with previous observations of neighborhood discrepancies in antisocial behavior (including crime), prosocial behavior, and social capital (Nettle, Colléony & Cockerill, 2011). Participants in Neighborhood B were far more likely than those in A to think that more of their neighbors behave uncooperatively. We could not, however, attribute this to a neighborhood difference in injunctive cooperative norms. Thus, a perceived lack of adherence to injunctive cooperative norms was pervasive in Neighborhood B.

This general perception in Neighborhood B that others are behaving antisocially appears justifiable: participants in Neighborhood B stole more money in the game. However, the results of our analyses suggest that this neighborhood difference in theft in the game can be explained by neighborhood differences in descriptive cooperative norms. That is, individuals who perceived cheating to be common were more likely to steal, and stole more in the game. These individuals tended to reside in Neighborhood B. Thus, the perception that others in the community are cheating may have induced further antisocial behavior in the game. While this observation is purely correlational, it is in accordance with the experimental results of Falk & Fischbacher (2002), who demonstrated a positive effect of observed theft on a participant’s subsequent choice to steal in the lab. It is also in agreement with those of Cialdini, Reno & Kallgren (1990) and Keizer, Lindenberg & Steg (2008), who showed that observed norm violation can result in an increase in norm violation.

Correspondingly, participants in Neighborhood B indicated far less trust in their neighbors than did those in A. This result fits with the far lower self-reported social capital in Neighborhood B previously observed. Our measure of trust in the current study, trust neighbors, approximates one of six items in the social capital index of Nettle, Colléony & Cockerill (2011), which was highly positively correlated with the overall index (0.77, p-value < 0.05).

As expected, and in concordance with social disorganization theory, trust neighbors was a positive predictor of punitiveness. Kocher, Martinsson & Visser (2012) similarly found that trust in members of a participant pool was positively correlated with punitiveness in a public goods game. Although they interpreted this outcome as stemming from greater disappointment in free-riding behavior, they suggest it merits further investigation of the role of social capital in norm enforcement.

One possible interpretation of the unpredicted interaction we observed between trust neighbors and value £2 lies in consideration of the multiple ways in which the cost of punishing can vary for the punisher. We showed that participants were more punitive when value £2 was smaller. Punitiveness is also less costly when there are fewer defectors and/or more punishers (Boyd et al., 2003, Gürerk, Irlenbusch & Rockenbach, 2006; Boyd, Gintis & Bowles, 2010). Trust neighbors may be informative as to whether Player 3 thinks there are many punishers and defectors in her neighborhood and thus construed as a measurement of the cost of intervening in antisocial behavior. From this perspective, our results are consistent with the idea that people are more punitive when punishment is cheap—with respect to both material resources and the behavior of others. This also highlights a limitation of this study, which is that Player 3 was able to punish anonymously and therefore ‘cheaply’ with respect to possible retribution. In the real world, third-party punishment may be associated with risk of retribution or other costs that are not captured by the £2 Player 3 paid to exact punishment. Decreased resiliency to retribution could also vary by neighborhood, perhaps partly as a result of differing material resources.

We are unable to determine whether participants in Neighborhood B stole more money than those in Neighborhood A because they thought punishment was less likely. This is because a participant’s motivation to steal a particular amount of money can be ascribed to inequity aversion as well as the expected probability of punishment. However, our data from Player 2 address expectation of punishment. While we did not observe a robust neighborhood difference in expect 3PP, we did observe a strong negative relationship between perceived cheating and expect 3PP. That is, a participant who thought many of her neighbors cheat on public goods was less likely to expect a neighbor to pay £2 to fine Player 1 if Player 1 took half her money.

This result supports the idea that descriptive cooperative norms are indeed informative as to expectation of punishment (Traxler & Winter, 2012). It also suggests that expectation of punishment is one of the mechanisms by which signs of norm violation can lead to further violation (Traxler & Winter, 2012; Kelling & Wilson, 1982). However, the causality of the observed relationship between perceived cheating and expect 3PP remains unknown. Surveys of the kind in Study 1 can only establish correlation; examining the causal significance of one variable for another requires experimental manipulation of the first variable. With this in mind, we undertook Study 2, in which we used selective feedback from Study 1 to experimentally alter perceptions of perceived cheating in the two neighborhoods.

Study 2

Feedback on or manipulation of descriptive norms has been used to alter people’s behavior—in diverse domains from littering (Cialdini, Reno & Kallgren, 1990) to energy use (Nolan et al., 2008). In Study 2, we used a novel method for manipulation of descriptive norms to investigate the causality of the relationship between perceived cheating and expect 3PP. In each neighborhood, we provided novice Player 2s with information on what their neighbors thought about the descriptive cooperative norms of the neighborhood (‘Norms treatment’). We manipulated this information so as to present Study 2 participants from Neighborhood A with a less positive picture of descriptive norms than was really the case, and participants from Neighborhood B with a more positive picture. We predicted that participants in Neighborhood A who received the Norms treatment would be less likely to expect Player 3 to 3PP on their behalf, compared to those participants in the same neighborhood who did not receive the treatment. We predicted the opposite effect in Neighborhood B.

Study 2 methods

Sampling

We collected data for Study 2 from October to December 2012, while Study 1 was ongoing (Supplemental Information), following the same protocol as in Study 1.

Norms questionnaire

We refer to the questionnaire used in Study 1 as ‘Baseline treatment’. The questionnaire for the Norms treatment differed as follows.

Civic norms manipulation: perceived cheating

The Norms questionnaire did not include questions about injunctive and descriptive norms. We presented participants with information on the responses of a subset of Study 1 participants in their neighborhood to the questions about descriptive civic norms (Supplemental Information).

The following backstory was used: as a part of the Tyneside Neighbourhoods Project, we had asked 10 people in their neighborhood how common they think avoiding a public transport fare, cheating the benefits system, and cheating on taxes, are in that neighborhood. We averaged these answers to get an idea of how common people think certain behaviors are. We wanted to know what other people in the neighborhood thought of these answers, and thus were asking them (Supplemental Information).

We presented one scale for each of the behaviors. The information in each scale was manipulated: in Neighborhood A, we took the mean of the 10 responses that gave the least favorable impression of cheating (i.e., high perceived cheating), and in Neighborhood B, we took the mean of the 10 responses that gave the most favorable impression of cheating (i.e., low perceived cheating). The information presented for Neighborhood A was: 5.7 for avoid a fare on public transport, 5.5 for cheat the benefits system, and 6.7 for cheat taxes (where 1 = ‘No one would’ and 10 = ‘Everyone would’). In Neighborhood B the information presented was: 2.2 for avoid a fare, 2.3 for cheat benefits, and 1.7 for cheat taxes. Beneath each scale, Study 2 participants were asked to circle ‘Fewer people would do this’, ‘This is about right’, or ‘More people would do this’ (Supplemental Information).

Contamination

To assess whether participants knew Study 1 participants, we included a contamination question: ‘Do you know of other people in your neighborhood who got a questionnaire and plan to post it or already have posted it?’ (‘Yes’, ‘Not sure’, or ‘No’).

3PP game

For Study 2, we measured the following behavior: expect 3PP (yes or no; representing whether Player 2 expected Player 3 to punish Player 1 if Player 1 took £5 from her).

Statistical analyses

We used binary logistic regression to assess the effect of the Norms treatment on expect 3PP within each neighborhood.

Study 2 results

Participants

For Study 2, we sampled 41 participants from Neighborhood A (21 male) and 39 participants from B (16 male) (Table S2).

Reaction to normative information

Participants in Neighborhood B were far more likely than those in A to indicate ‘This is about right’ when presented with the manipulated norms scales for cheat benefits and avoid fare (OR 3.63, 95% CI [1.23–10.70] and OR 3.74, 95% CI [1.34–10.49], respectively). In Neighborhood B, 38.46%, 43.59%, and 46.15% of participants indicated ‘This is about right’ for cheat benefits, avoid fare, and cheat taxes, respectively. In contrast, the majority of participants in Neighborhood A indicated ‘Fewer people would do this’ when presented with the manipulated scales for cheat benefits and avoid fare (78.05% of participants for each behavior). Only 51.28% of participants in Neighborhood A indicated ‘Fewer people would do this’ for cheat taxes.

Expectation of 3PP: norms treatment

Participants in Neighborhood B who received the Norms treatment—i.e., who received information that their neighbors perceive cheating to be uncommon—were more likely to expect Player 3 to 3PP on their behalf, compared to those in B who received the Baseline treatment. The proportion of participants who expected 3PP is 58.97% for the Norms treatment, compared to 30.23% for Baseline (OR 3.32, 95% CI [1.33–8.25]; Fig. 6). Exclusion of participants for whom contamination was ‘Not sure’ (five) or ‘Yes’ (two) does not qualitatively change the results. (One participant circled both.)

Figure 6 Proportion of Player 2s, by neighborhood and treatment, who indicated that they expect Player 3 to 3PP on their behalf.

We did not observe a robust effect of the Norms treatment on expect 3PP in Neighborhood A. Contrary to our prediction, the proportion of participants in A who expected 3PP is 41.46% for Norms treatment, compared to 36.36% for Baseline treatment (OR 1.24, 95% CI [0.52–3.00]; Fig. 6).

However, the Norms treatment generated an unanticipated response in Neighborhood A. Some participants attempted to redirect their money by asking us to: donate it to charity (three participants), keep it for research/university funds (two participants), or not pay them (one participant). The rate of ‘opting out of payment’ is 14.63% for Norms treatment participants in Neighborhood A, compared to 1.15% of Baseline participants in A (OR 11.25, 95% CI [2.18–57.97]). This spontaneous change in game play was not observed in Neighborhood B.

Study 2 summary and discussion

In Study 2, participants in Neighborhood B received information that their neighbors think there is little cheating on public goods in their neighborhood, relative to what we actually observed in Study 1. They were far more likely to expect a neighbor to punish antisocial behavior compared to those in Neighborhood B who did not receive the manipulation. Whether disorder can play a causal role in an increase in crime rates (Kelling & Wilson, 1982) has been debated (Sampson, Morenoff & Gannon-Rowley, 2002; Markowitz et al., 2001). Our results provide empirical evidence of a mechanism by which norm violation can lead to the further violation of a different norm—through change in the expectation of punishment.

There are at least three plausible routes by which this effect is achieved. One possibility is that people expect cooperators to be more likely than non-cooperators to punish. The second is that people perceive other’s behavior to reflect other’s expectation of punishment. That is, people think that others are not behaving antisocially because of their expectations of punishment for behaving antisocially. The third possibility, closely related to the second, is that if antisocial behavior is very common, people may intuit that it persists because antisocial behavior is going unpunished and thus have a decreased expectation of punishment.

We did not observe a reliable negative effect of the norms manipulation on expectation of 3PP in Neighborhood A. It is not clear why we observed the expected result in Neighborhood B and not Neighborhood A. In Study 1, we found greater variation in trust and norms in Neighborhood B than in Neighborhood A (Table 1). One interpretation of this is that the environment is more heterogeneous and unpredictable in Neighborhood B. If so, perhaps residents of Neighborhood B are less certain than residents of Neighborhood A of the behavior of their neighbors and therefore were more accepting of the manipulation. Indeed, far more Neighborhood B participants circled ‘This is about right’ when presented with the manipulated descriptive norms. Another possibility is that participants in Neighborhood B were more accepting of the information provided by an authority figure (university personnel/scientist).

General Discussion

The aim of this paper was to consider how the local social environment affects individual decisions to engage in and sanction antisocial behavior, and how an individual’s antisocial behavior can in turn affect the local social environment, by conveying information about descriptive norms. In Study 1, we observed that subjects in Neighborhood B took more money from their neighbors and were less punitive in an economic game of crime and punishment. The perception that others are cheating on public goods varied dramatically by neighborhood, was fundamental to the neighborhood difference in theft in the game, and was negatively associated with the expectation of third party punishment for antisocial behavior. Subjects in Neighborhood B were also less punitive of antisocial behavior, and punitiveness was negatively associated with trust in one’s neighbors.

In Study 2, we showed that providing participants in Neighborhood B with information that cheating is perceived as uncommon within their neighborhood led to a sharp increase in the expectation of third-party punishment for theft. An increase in the perceived likelihood of punishment would presumably lead to greater cooperation, given the close relationship between these two variables. Thus, these results provide novel empirical support for a mechanism by which cues of norm violation can lead to further norm violation (Cialdini, Reno & Kallgren, 1990; Keizer, Lindenberg & Steg, 2008): altered expectation of punishment (Kelling & Wilson, 1982; Traxler & Winter, 2012).

We consider these results within a framework where culture is dynamic, subject to evolutionary processes that can lead to more or less cooperative outcomes (Boyd & Richerson, 1985). Unlike in recent cross-cultural studies of cooperation and punishment (Henrich et al., 2006; Herrmann, Thöni & Gächter, 2008), our two study populations share many cultural components, including the institutions that formally sanction their civic violations (although how those institutions are experienced may vary) and injunctive cooperative norms. The apparently large discrepancy between desired and achieved cooperative outcomes in Neighborhood B, as assessed with injunctive and descriptive cooperative norms, adds a new perspective on the cultural evolution of variable cooperative outcomes.

Our results provide evidence for three potential routes by which perceived cooperative norm violation can lead to further violation of cooperative norms.1 All of these processes have been postulated or investigated by others; however, to our knowledge, they have not been considered simultaneously as processes that may, in concert, lead to substantial cultural change. These positive feedback processes are: (1) To avoid being ‘suckered’, conditional cooperators are motivated to defect if they perceive that defection is common (Fischbacher, Gächter & Fehr, 2001; Falk & Fischbacher, 2002; Bichierri & Xiao, 2009; Raihani & Hart, 2010; Irwin & Simpson, 2013). (2) Perceived cheating leads to lower trust. Low trust results in reduced informal punishment of norm violation (Kocher, Martinsson & Visser, 2012). In this vein, Traxler & Winter (2012) observe a direct effect of the perceived frequency of norm violations on expressed willingness to sanction violations. Similarly, extensions of social disorganization theory include feedback processes between crime/disorder and social cohesion/control, via fear or residential instability (Sampson & Raudenbush, 1999; Markowitz et al., 2001; Steenbeck & Hipp, 2011). (3) When the perceived frequency of cooperative norm violation is high, expectation of punishment for violation is lower (Sah, 1991; Traxler & Winter, 2012).

We hypothesize that these three positive feedback mechanisms, wherein perceived cooperative norm violation leads to further cooperative norm violation, could act simultaneously to result in a rapid downward spiral, leading to low levels of cooperation. As Cialdini, Reno & Kallgren (1990) note, descriptive norms are informative as to adaptive behavior. In a community with low levels of cooperation and minimal punishment of cooperative norm violation, non-cooperative strategies may outperform others (Wilson & Csikszentmihalyi, 2007). Other processes—prestige-biased (Henrich & Gil-White, 2001) or conformist (Henrich & Boyd, 1998) transmission and self-selection of people with preferences for an antisocial community—could further reinforce uncooperative or overtly antisocial strategies. While cooperative norms are considered a component of social capital (Knack & Keefer, 1997; Bowles & Gintis, 2002), our results demonstrate the need for explicit integration of cultural transmission and norm psychology—i.e., psychological adaptations for determining and adopting local norms and punishing violators (Chudek & Henrich, 2011)—with social disorganization theory. Scholars of criminology will note some similarities between the social learning theory of deviance (Akers, 2009) and theories of cultural transmission. However, we extend this bridge between the social environment and individual behavior by emphasizing the feedback from the individual to the social group. That is, we have outlined three routes by which an individual’s defection can lead other individuals to adopt similar behavioral strategies, thus altering the local cultural ecology (Camerer & Fehr, 2006).

Missing from this hypothesized downward spiral is an initial perturbation that could result in an increase in cooperative norm violation (or perceived violation) in the neighborhood. Poverty and economic uncertainty are also striking differences between Neighborhoods A and B. Without middle class buffers of savings and credit, institutional safety nets, or strong reciprocal networks, crises such as illness create the potential for dire outcomes, thus altering the costs and benefits of defecting. For people already living at the margin, material crises might result in a higher probability of defection. Especially for crises that hit broad swaths of a community simultaneously, such as the widespread job loss in Neighborhood B resulting from the collapse of the shipbuilding and coal mining industries, one can imagine an increase in the frequency of defection that alters the descriptive cooperative norms enough to start a downward spiral in defection.

Importantly, although we hypothesize that poverty and economic uncertainty were linked to an initial perturbation of cooperative norm violation in the current study, the positive feedback of norm violation could continue in the absence of poverty. There has been debate as to whether there are direct, as well as indirect, effects of poverty and/or income inequality on crime (Patterson, 1991; Bursik & Grasmick, 1993a; Bursik & Grasmick, 1993b). The story we have sketched is compatible with both possibilities, as an historical direct effect of poverty on norm violation may lead to cultural dynamics that persist beyond the duration of the poverty itself. (For a similar example of such cultural inertia, see Sah (1991), who argues that a transient change in the economics of crime can lead to persistently high crime rates, due to a postulated relationship between higher crime rates and decreased expectation of punishment.)

However, we can only speculate as to whether these dynamics are at play in Neighborhood B (outside of the 3PP game) and to what extent they can explain the observed high rates of crime and antisocial behavior.

This paper also makes contributions to empirical gaps in two fields. In Study 1, we demonstrated that the covariation of cooperation and punishment of non-cooperation, which has been observed cross-culturally with economic games (Henrich et al., 2006), can extend to the local level. Participants in Neighborhood A stole less money and were more punitive in the game than those in B. Also in Study 1, we demonstrated an association—albeit small—between third-party punishment of antisocial behavior and trust in one’s neighbors, as well as a neighborhood-level association between antisocial behavior in the game and decreased third-party punishment of antisocial behavior. These results provides additional, novel empirical support for the relationship between (1) low social control and low social capital, and (2) low social control and high rates of antisocial behavior. Data on actual social control (rather than the potential that residents will engage in social control, as measured by survey data) are difficult to come by, limiting the strength of the inference that low social capital and high rates of antisocial behavior are correlated due to lack of social control (Bursik & Grasmick, 1993a; Bursik & Grasmick, 1993b; Steenbeck & Hipp, 2011).

We acknowledge that there are a number of limitations to our studies. We could not control the order at which participants looked at or filled out packet components. It is possible that participants ‘justified’ their behavior in the game with their questionnaire answers. However, we might then expect a robust positive effect of value £1 on theft. Presenting Player 1s with the threat of punishment for theft could have decreased intrinsic motivation to behave cooperatively (Frey & Jegen, 2001), although it is unclear how this would produce a spurious correlation between perceived cheating and theft in the game. We cannot account for the neighborhood residents who chose not to respond, although in both neighborhoods we likely reached a segment of the community biased towards prosocial preferences (registered voters and research participants). Additionally, although participants were anonymous to each other in the game, they were not anonymous to us. The neighborhood differences in game behavior we observed could be partly attributed to participants in Neighborhood A, but not Neighborhood B, regarding a university professor as someone in their social milieu and thus being concerned about reputational repercussions.

Finally, we have two related suggestions for future study that may increase our understanding of why some communities appear to be stuck at uncooperative equilibria, despite concerted efforts by city planners to chart a different course (Robinson, 2005), or even substantial temporal changes in the demographic makeup (Shaw & McKay, 1942). The first is further investigation of the potential for multiple, simultaneous paths of positive feedback on cooperative norm violation, including not just conditional cooperation but also punitiveness and expectation of punishment. The second is consideration of how psychological adaptations for recognizing and adopting local norms, as well as biased in- and out-migration (Chudek & Henrich, 2011), can reinforce an antisocial culture.

Supplemental Information

Supplemental Information 1 Supplemental text and tables and a copy of the questionnaires

Click here for additional data file.

We thank residents of Newcastle Upon Tyne for participating in these studies, and we thank MN Grote, R McElreath, and K Rauch for helpful discussion and comments.

Additional Information and Declarations

Competing Interests

Author Contributions

Human Ethics

1 Our studies focused on cooperative norm violation (“perceived cheating”) and a very specific type of antisocial behavior (theft). However, based on work on cross-norm effects referenced in the introduction, we think that we can draw inferences here not just about theft but cooperative norm violation in general.

The authors declare there are no competing interests.

Kari Britt Schroeder conceived and designed the experiments, performed the experiments, analyzed the data, wrote the paper, prepared figures and/or tables, reviewed drafts of the paper.

Gillian V. Pepper and Daniel Nettle conceived and designed the experiments, performed the experiments, reviewed drafts of the paper.

The following information was supplied relating to ethical approvals (i.e., approving body and any reference numbers):

The Ethics Committee of the Newcastle University Faculty of Medical Sciences approved the study protocol: Protocol #00503/2011 and Amendment #00503_1/2012.

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
