# Peer review of "Local norms of cheating and the cultural evolution of crime and punishment: a study of two urban neighborhoods"

_PeerJ, doi:10.7717/peerj.450_

## Round 0.1 · original submission · Minor Revisions

This is a very interesting paper however as pointed out by one of the reviewers it can be confusing in places; some sections I had to reread several times. Where possible please try to simplify the text although I appreciate that this may be easier said than done.

Reviewer 1 ·

Basic reporting

A solid, well-written paper

Experimental design

Fine

Validity of the findings

The findings seem sound---the samples are sufficient and the analysis is sophisticated.

Additional comments

This paper reports both observations and experiments in two neighborhoods in Newcastle on Tyne aimed at understanding the effect of norms on behavior and behavior on norms. It is a very well done study. The experiments are well designed, the authors have a very good grasp of the relationship of these results to the relevant theory in evolutionary psychology, and the results are very interesting. I strongly recommend publication.
I do have one question/suggestion. We are told nothing about the residents of these two neighborhoods beyond income and the results of the questionnaires. I’m not familiar with Newcastle on Tyne, but I would have thought that most British cities would have significant variation in ethnicity. If there is such, it would be interesting to look at individual ethnicity as a predictors, and if possible, an interaction of ethnicity with number of generations resident in the UK. In the US, ethnicity sometimes predicts political/social attitudes for several generations. If such data was collected, it would be interesting to use one of the corruption indices for countries of origins as a covariate as well.
Here a few minor comments keyed to line number:
156 Did you ask before or after player 2 had committed to level of theft? If before, this seems like it might have created a demand effect---make it more likely player two considers the possibility that player 3 will punish.
307 This interaction could be due to nonmonetary costs that correlate with income. So individuals who value money more are also more vulnerable to payback.
412 I’m not real keen on the use of the 3PP acronym as a verb.

·

Basic reporting

No comments.

Experimental design

The experimental design is generally clever and strong.

Validity of the findings

The fundamental validity of the findings is defensible, however I have some concern about their interpretation--which is a form of validity in itself. See the General Comments below.

Additional comments

This reviews the manuscript “Local norms of cheating and the cultural evolution of crime and punishment,” submitted for consideration for publication at PeerJ. The manuscript presents two studies that use experimental economics games to compare social behavior between two neighborhoods in Newcastle Upon Tyne, England. They use a third-party punishment (3PP) game in conjunction with survey measures to examine the relationship between an individual’s perceptions of their neighbors’ social tendencies, and how this then influences the manner in which the same individual approaches the game. The first study found that individuals who perceived their neighbors as being less trustworthy also expected less third-party punishment to occur in the game. The second study then manipulated perceptions of trustworthiness through an experimental prime. In the neighborhood with less trust (and less cooperation), the presentation of information that neighbors are in fact trustworthy led to a greater expectation of third-party punishment. This further validated the findings of the first study.
I appreciate the overall thrust and intent of this manuscript, as I have enjoyed the series of studies preceding it from the work of Nettle and his colleagues in the Newcastle Upon Tyne project. This work has been an elegant way to leverage the ecological diversity of urban spaces, and I admire the ambition of the methodology utilized here. I have had a lot of difficulty working through the manuscript, however, and I feel as though the authors have had a similar amount of difficulty in identifying the contribution that they are making, and the manner in which they want to present it. I would like to address this across a handful of themes three themes: interdisciplinary theorizing; interpretive clarity; clear presentation.

Interdisciplinary theorizing
Most critically, the authors have written a robustly interdisciplinary paper, spanning “the behavioral, economic, evolutionary, sociological, and social psychological literatures” (lines 53-54), but the conceptual intersection is not entirely clear. It feels more like ideas from these disparate fields have been stitched together to address the specific data here, and sometimes in ways that are not entirely compelling. For example, the first two paragraphs connect evolutionary perspectives on cooperation to social disorganization theory by equating the creation of a crime-free neighborhood to the contribution to a public good. This is not an inappropriate comparison, but one that would require quite a bit of theoretical development that is missing—social disorganization theorists think about the crime-free neighborhood as a consequence of collective efficacy in the production of public order, and that process of producing order might then be seen as a sort of abstract public good that relies on the individual and coordinated actions of local residents. Again, the comparison is not wrong, but assuming that these two audiences will readily see legitimacy in it can be tricky.

With that said, I’d like to recommend that the authors focus the manuscript more tightly around norm enforcement. This is a concept seen as focal to both fields and where the terminology is about the same. They both respect Cialdini and Keizer’s work, for example. The connection to broken windows theory and the cycle of disorder and decline might be useful as well, but it is not immediately apparent to me that it is essential, as I discuss in the next theme. In any case, these need to be woven together into a narrative that is more generally accessible, provided that that general audience is what the authors are looking for.

Interpretive Clarity
I am a bit uncertain about what has been discovered here. I think I get it in strict empirical terms, but I am not quite sure what it actually means. I would not have predicted that my perception of civic behaviors among my neighborhoods would predict my expectation for them to punish other people who are being antisocial, and I don’t think Cialdini-esque arguments or Keizer’s more recent model of norm violation necessarily would predict it either, in large part because my perceptions of others aren’t affecting my behavior—they are instead affecting my assumptions about their behavior in an unfamiliar task. Further, most research—be it evolutionary, behavioral economic, or sociological—think of cooperation and norm enforcement as conceptually distinct behaviors. While they have been shown to correlate at the group level, they have somewhat different motivations and therefore correlations at the individual level. So, for my belief in the civicness of my neighbors to influence my belief in their punishment behavior does not automatically follow the theory discussed.

Similarly, there is the proposal of a broken windows-like downward spiral. But the results are not showing specifically how perceptions of others are driving behavior. Their point 3 (Lines 464-477), as I have just addressed, is how perceptions in one domain are transferred over to another domain. I think stating this as a broken windows situation is a little bit of a stretch in that case. Point 1 is much stronger. Point 2 is that perceived cheating leads to lower trust which leads to a lower likelihood of punishing. However, it is not clear with the interaction effect that this is the case—it seems to be largely dependent as well on the amount that punishment would subjectively cost the participant. Altogether, this leaves 1 of 3 arguments on steady ground.

To sum up, there are a lot of findings, but I’m not sure what the story is.

Clear Presentation

In a similar vein, I find the methodology and results themselves difficult to navigate. I think this is a byproduct of the complexity of the experimental economics protocols, which is only exaggerated by the neighborhood-based experimental design and additional survey questions. I appreciate this difficulty, but I think some effort needs to be taken to make the manuscript more navigable. The answer may be dropping some analytical tests, focusing in on those that “tell a story” and excluding those that are not relevant. This would require a somewhat different tack in the introduction and presentation of the hypotheses, but might strengthen the contribution of the manuscript considerably.

The actual analyses might be revised as well to be a bit more direct in what they are testing and finding. I found myself rereading sections to make sure I understood them. Last, referring to measures consistently as their italicized variable name seems unnecessary. In many cases, the name on its own creates a syntactically awkward sentence.

Small points:
--Lines 40-41: Social disorganization theorists state that poverty, residential mobility, and family disruption can *mitigate the capacity a community has for creating relationships and establishing shared social norms*, which then results in reduced participation, etc.
--Does the statistical test used throughout the manuscript offer p-values? If so, it would be good to report them.

I hope that these comments and suggestions are helpful to the authors in preparing a revised manuscript.

---

## Round 0.2 · accepted · Accept

Thanks you for the clear response to the reviewers and for making a paper a lot clearer. I must say I found this a fascinating paper and look forward to seeing in published.